# Differential Features of Fusion Activation within the *Paramyxoviridae*

**DOI:** 10.3390/v12020161

**Published:** 2020-01-30

**Authors:** Kristopher D. Azarm, Benhur Lee

**Affiliations:** Icahn School of Medicine at Mount Sinai, New York, NY 10029, USA; kristopher.azarm@icahn.mssm.edu

**Keywords:** paramyxovirus, viral envelope proteins, type I fusion protein, henipavirus, virus entry, viral transmission, structure, rubulavirus, parainfluenza virus

## Abstract

Paramyxovirus (PMV) entry requires the coordinated action of two envelope glycoproteins, the receptor binding protein (RBP) and fusion protein (F). The sequence of events that occurs during the PMV entry process is tightly regulated. This regulation ensures entry will only initiate when the virion is in the vicinity of a target cell membrane. Here, we review recent structural and mechanistic studies to delineate the entry features that are shared and distinct amongst the *Paramyxoviridae*. In general, we observe overarching distinctions between the protein-using RBPs and the sialic acid- (SA-) using RBPs, including how their stalk domains differentially trigger F. Moreover, through sequence comparisons, we identify greater structural and functional conservation amongst the PMV fusion proteins, as compared to the RBPs. When examining the relative contributions to sequence conservation of the globular head versus stalk domains of the RBP, we observe that, for the protein-using PMVs, the stalk domains exhibit higher conservation and find the opposite trend is true for SA-using PMVs. A better understanding of conserved and distinct features that govern the entry of protein-using versus SA-using PMVs will inform the rational design of broader spectrum therapeutics that impede this process.

## 1. Introduction

The family *Paramyxoviridae* (Order: *Mononegavirales*, Class: *Monjiviricetes*, Subphylum: *Haploviricotina*, Phylum: *Negarnaviricotina*, Realm: *Riboviria*) comprises of enveloped, non-segmented negative-sense RNA viruses that have a vast host range, ranging from mammals and birds to reptiles and fish [1]. The most recent classification of *Paramyxoviridae* by the International Committee on Taxonomy of Viruses (ICTV) includes four subfamilies, 14 genera, and 72 recognized species (Figure 1). Paramyxoviruses (PMVs) belonging to divergent genera such as measles virus (*Morbillivirus*), mumps virus (*Orthorubulavirus*), the human parainfluenza viruses (*Orthorubulavirus*, *Respirovirus*), and Nipah and Hendra viruses (*Henipavirus*) are major human-tropic pathogens of global biomedical importance. Human-to-human transmission of measles, mumps and the parainfluenza viruses likely occurs through airborne routes or fomites [2], while direct contact with infectious bodily fluids (respiratory secretions, saliva, urine) is required for transmission of henipaviruses (HNVs) [3,4,5]. HNVs, including the zoonotic Hendra and Nipah viruses (HeV and NiV), are highly pathogenic in humans and domestic animals when spillover occurs from their natural chiropteran (bat) hosts. While HeV outbreaks appear confined to northeastern Australia, NiV outbreaks occur with alarming regularity in South/Southeast Asia and result in mortality rates upward of 90% [5,6,7]. More recently, sampling of bat reservoirs in parts of South America and Africa—areas distinct from what was thought to be the normal geographical distribution for extant HNVs [8,9,10]—has not only shown evidence for divergent HNV species, but has also revealed that bats host PMVs from major genera (e.g., morbilliviruses, respiroviruses, orthorubulaviruses and pararubulaviruses) that comprise of species pathogenic to humans or domestic animals. Furthermore, serological evidence suggests the divergent HNVs in Africa have previously spilled over into the human population [11]. Thus, both extant and emerging PMVs pose a threat to global health. 

Paramyxovirus (PMV) entry occurs via pH-independent fusion of the viral envelope with the host cell membrane. For the majority of PMVs, this process involves the concerted actions of two viral glycoproteins: the receptor-binding protein (RBP) and fusion protein (F). There are various flavors of RBP [12,13,14,15]. HN, hemagglutinin-neuraminidase, proteins bind sialic acid (SA) on red blood cells (RBCs) and cause them to aggregate (hemagglutination activity), in addition to being able to cleave SA (neuraminidase activity) to facilitate virion release from the infected cell [16,17,18,19,20,21]. Hemagglutinin (H) proteins of the genus *Morbillivirus* solely have hemagglutination activity, aggregating RBCs from certain primates, but not humans, by binding to CD46 [22]. For systemic spread and respiratory transmission in humans, MeV-RBP uses the physiologically relevant SLAM (or CD150) and Nectin-4 proteins, respectively [23,24,25,26,27]. Finally, remaining members of the *Orthoparamyxovirinae*, that do not express HN or H proteins, possess G proteins that are either experimentally confirmed or predicted to bind proteinaceous receptors [28,29,30,31,32,33]. More recently, the RBP from members of the *Pararubulavirus* genera (e.g., Sosuga virus), presumed to contain HN activity, has been shown to not use SA-based receptors [34] (Figure 2 and below for more detailed discussion). The RBP binds the host receptor on the target cell. Following receptor binding, the RBP undergoes a conformational change that allosterically triggers the metastable fusion protein, which then undergoes its own conformational cascade that eventually facilitates the merging of the viral envelope with the host cell membrane. Compared to the rest of phylum *Negarnaviricota*, the necessity of (at least) two viral proteins to enact entry is unique to paramyxoviruses. Although weak fusion triggering in the absence of the RBP [35,36] was reported for parainfluenza virus 5 (PIV5) and Sendai virus (SeV), it is unclear if these are physiologically relevant phenomena. Upon infection, the cell expresses the RBP and F protein on its surface and fuses with naive receptor-expressing cells, forming multi-nucleated cells (syncytia) [13,37,38,39,40]. Depending on the cell type and virus, PMVs can spread more efficiently either via cell-free viral particles or direct cell-cell spread [41,42,43].

Understanding the mechanisms of PMV entry allows for the identification of vulnerable targets on the virus. These findings inform the rational design of therapeutics that bind these targets and render the virus unable to infect. There have been many excellent reviews on paramyxovirus entry [13,14,15,44,45,46]. This particular review highlights the PMV entry pathway, focusing specifically on the functional differences in fusion activation between members of the family.

## 2. Modes of Receptor Binding

Binding of the viral attachment glycoprotein, now formally designated as the receptor binding protein (RBP) [1], to the host cell receptor is the first step in paramyxovirus (PMV) entry. Receptor binding occurs on the globular head domain of the RBP, which folds as a six-bladed *β*-propeller [32,33,47,48,49,50,51,52,53,54,55,56]. There are two major classes of paramyxoviral RBPs: (1) those that bind to sialic acid-containing surface molecules (e.g., viruses that comprise the *Orthorubulavirus*, *Orthoavulavirus*, and *Respirovirus* genera) [57,58] or (2) those that bind to proteinaceous receptors (e.g., viruses belonging to *Morbillivirus* [23,24,25,26,59,60,61] and *Henipavirus* [28,29,62] genera). 

### 2.1. SA-Using PMVs

Avula-, respiro-, rubula-, ferla- and aquaparamyxo- viruses express RBPs that recognize sialic acids (SAs) on glycoproteins and glycolipids. Another essential function of their RBPs is neuraminidase (NA) or SA-cleavage activity, which releases the budding virion from the parent cell and prevents reinfection of the same cell. It has recently been observed that several members of the *Pararubulavirus* genus lack the essential neuraminidase activity motifs in their RBPs [1,34]. These motifs include the arginine triad (Arg174, Arg416, and Arg498 for NDV) and an “Asn-Arg-Lys-Ser-Cys-Ser” hexapeptide motif, both of which have been shown to be required for binding and hydrolyzing SAs (Figure 2) [63,64,65]. The absence of these conserved residues indicates that pararubulaviruses could use protein receptors, functionally distinguishing them from the genetically related SA-using orthorubulaviruses. Indeed, Stelfox et al. (2019) provided evidence that Sosuga virus (SosV) likely does not use SA for entry [34]; SosV-RBP does not possess the conserved hexapeptide motif (Figure 2), does not exhibit HN functionality, and structural analysis indicates that its six-bladed *β*-propeller globular head domain is incompatible with SA binding.

The two major human pathogens of the SA-using paramyxoviruses include human parainfluenza virus type 3 (HPIV3), the clinically most prevalent HPIV subtype, and mumps virus (MuV). Both HPIV3- and MuV-RBP (HN) preferentially recognize α2,3-linked sialic acids in branched and unbranched oligosaccharides present on either glycoproteins or glycolipids [66,67]. However, some MuV strains, especially neurovirulent variants, may show increased binding activity for α2,6-linked sialic acid, the linkage present on nervous tissues [68]. These results suggest SA-linkage preferences may influence viral pathogenesis. With respect to the SA-binding activity, the human paramyxoviruses HPIV3 and MuV differ from human influenza viruses, which have a clear preference for the α2,6 linkage [69]. Despite the classical view that α2,6 and α2,3 receptors are concentrated in the human upper (URT) and lower respiratory tract (LRT), respectively, recent glycomic analyses based on lectin binding and mass spectrometric methods have revealed non-localized distributions of α2,6 and α2,3 receptors in the respiratory tract [70,71,72]. Nonetheless, in general, the different HN proteins have different binding preferences to different SA-end glycans, potentially impacting systemic distribution [57,58].

### 2.2. Ephrin-B2-Using PMVs

All extant members of the *Henipavirus* (HNV) genus, with the exception of Mojiang virus (MojV), use the ephrin-B2 ligand for entry [28,29,30,73,74]. Ephrin-B2 is expressed in endothelial cells and in neurons, dictating the tissue tropism of these viruses during natural infection [37,75,76]. Alternative receptor usage of the closely related host ephrin-B3 is also reported for NiV and HeV [62,77]. Interestingly, Cedar virus (CedV) can use ephrin-B1 [32,33] in addition to ephrin-B2 on physiologically relevant primary endothelial cells [33]. 

The putative rat-borne henipavirus, MojV, appears not to use any of the canonical paramyxovirus entry receptors and possesses a structurally distinct RBP [31]. Thus, the discovery and emergence of novel paramyxoviruses that use idiosyncratic entry pathways underscore the diversity in host receptors this genus has evolved to exploit. 

### 2.3. CD150-/Nectin-4-/CD46-Using PMVs

Measles virus RBP (MeV-RBP) uses three proteinaceous cellular receptors depending on the virus strain and tissue type: SLAM (or CD150), CD46 (only for laboratory/vaccine strains), and Nectin-4 [23,24,25,27,54,61,78,79]. 

MeV-RBP (formerly MeV-H) receptor binding occurs on a unique face of the *β*-propeller. For HPIV3-RBP (HN) and NiV-RBP (G), receptor binding takes place at the central core of the six-bladed *β*-propeller (Figure 3A,B), whereas the receptor binding surface for MeV-RBP maps to the “side” of the *β*-propeller (Figure 3C) [50,51,54]. This distinction and structural phylogeny analysis [31] indicate that NiV-RBP is structurally more closely related to HPIV3-RBP than MeV-RBP, and suggest that “HN” may be the ancestral RBP from which H and G RBPs independently arose [56].

Moreover, the binding sites for the three MeV-H receptors are only partially overlapping on the side of the six bladed *β*-propeller head, but there is a stronger overlap between the footprints of Nectin-4 and CD46 (Figure 3C). Considering CD46 usage developed entirely as a result of growth in cell culture, the interface overlap suggests the CD46 binding site on the RBP of vaccine strains may have arisen from the Nectin-4 binding site [80]. The identity of the “original” MeV-RBP receptor, either SLAM or Nectin-4, remains unknown.

## 3. Conformational Changes in the RBP

PMV RBPs are composed of a C-terminal globular head domain that folds as a six bladed *β*-propeller and an N-terminal 4-helix bundle (4HB) stalk domain. The extant atomic structures of the RBP (HN, H, or G) globular head domains reveal high structural conservation [48,50,51,52,53,54,56,82,83,84]. 

The RBPs show a dimer-of-dimers arrangement of the globular heads with oligomerization occurring through covalent or non-covalent associations in the stalk domain [48,49,85,86,87]. Moreover, the RBP stalks have either been crystallized as [55,88] or are modeled as 4HBs with strong central hydrophobic cores [89,90,91]. Representative extant or modeled stalk monomers are shown in Figure 4. The stalk domains of the protein-using PMVs are longer than the stalk domains of the SA-using PMVs (106 residues for NiV/HeV vs. 96 residues for MeV vs. 69 residues for PIV5 RBPs). However, based on the extant PMV-F atomic structures, the fusion proteins stand at similar heights [92,93,94,95,96,97,98]. The biological reasons for the relative height differences in the RBP are still unclear. 

### 3.1. Insights from the Crystal Structures of the 4HB

As observed in both the NDV-RBP and PIV5-RBP crystallized stalk domains, the 4HB structure for RBPs (HNs) is characterized by a buried hydrophobic core with each monomeric stalk domain adopting a left-handed supercoil 7mer repeat lower (membrane proximal) region that transitions to a non-supercoiled 11mer repeat upper (membrane distal) region (Figure 5A). A polar residue(s) marks the superhelical transition point (e.g., S79 for PIV5-RBP) from heptad to 11mer repeats and creates a kink in the 4HB (Figure 5B, Figure 4). The stalk domain drives tetramerization, as the isolated globular head domains exist primarily as monomers [48,82,99,100]. Canonically, HN dimers are linked by a disulfide bond in the stalk region [48,86,101] and form tetramers via noncovalent hydrophobic interactions [55,102]. Comparing the soluble RBP stalks, the PIV5 ectodomain exists as a homogenous tetramer in solution, whereas the NDV-RBP ectodomain exists as predominantly dimers [48,55,100]. Three polar residues disrupt the hydrophobic core of the NDV-RBP 4HB, whereas PIV5-RBP only has one such residue, explaining this relative oligomeric instability [88]. By contrast, henipaviruses form stable tetramers reinforced by three disulfide linkages in the stalk domain [90].

Two categories of stalk mutations have been identified for RBPs with HN functionality that each have a different impact on tetramerization of the 4HB [55]. Category I mutations affect only F activation, map to the outer surfaces of the 4HB, and disrupt RBP-F interactions. Category II mutations affect both NA and fusion-promoting functions and are buried in the hydrophobic core of the 4HB. For category II mutants, ultimate loss of the stalk structure and disassembly of the RBP tetramer likely perturbs NA activity. Given that category II mutations—which occur exclusively in the hydrophobic core—have an impact on tetramerization, whereas category I mutations—which occur on the exposed surface of the 4HB—do not, the differing phenotypes further support the solo involvement of the hydrophobic core in tetramerization. Moreover, category II mutations are linked to lowered pathogenicity, suggesting the tetrameric integrity of RBP plays a role in NDV pathogenesis [110]. 

### 3.2. Allosteric Changes upon Receptor Binding

There are only minor conformational differences between the apo- and receptor-bound structures of the six-bladed *β*-propeller globular head domains [48,52,53,56,83,111]. However, these renderings lack the structural contributions of the stalk domain. In both the NDV-RBP and PIV5-RBP ectodomain atomic structures, in which the 4HB stalk was captured, the globular heads of the tetramer were observed in either “heads up” or “heads down” positions. It was proposed that the globular head domains attain these various positions with respect to the stalk domains, depending on the presence or absence of receptor binding. Notably, the stalk residues that are obscured in the “heads down” conformation are those implicated in F activation, i.e., the stalk exposure mechanism discussed later in this text. 

Moreover, a recent study by Wong et al. (2017) found monomeric ephrin-B2 receptor binding induced allosteric changes in the NiV-RBP ectodomain distinct from that induced by dimeric ephrin-B2 [112] binding. These allosteric transitions were characterized by movement of the globular heads relative to the stalk. Importantly, using deuterium exchange (HDX-MS) methods, the stalk was identified as a dynamic region upon receptor binding, supporting the stalk exposure mechanism for fusion protein (F) triggering.

## 4. The Paramyxovirus Fusion Protein: The Energetic Facilitator of Fusion

The paramyxovirus fusion proteins (PMV-Fs) are trimeric class I viral membrane fusion proteins [93,94,95,96,113,114]. Each monomer of the homotrimer has N-(HR1) and C-(HR2) terminal heptad repeat domains, and a cleavage site located immediately N-terminal to the fusion peptide. The precursor F_0_ needs to be cleaved into the disulfide linked F_1_ and F_2_ subunits in order for the trimer to be biologically active. 

For most paramyxoviruses, the fusion protein is cleaved by cellular furin-like proteases in the trans Golgi network during cell surface trafficking or extracellular Clara cell-type tryptases present in the respiratory tract or embryonated egg allantoic fluid [115,116,117,118,119,120,121]. For HNVs, the premature F protein is endocytosed from the cell surface and cleaved by cathepsin L or cathepsin B in the endosome [122,123,124,125,126,127] and then re-cycled back to the cell surface.

The metastability of PMV-F ultimately drives the virus-host cell membrane merger (reviewed in [12,14,15]) (Figure 6). Following both protease cleavage and activating signals from the RBP, the HR1 domains of the trimer convert from a set of compacted helical structures and beta sheets into a highly unstable, but presumably structured, extended helical trimeric coiled-coil domain (Figure 6, Panel 2). This coiled-coil domain is terminated by three hydrophobic fusion peptides (FPs) that insert into the target membrane. Thus, the target membrane must be within the range of insertion of the FP–a distance of 100–210 Å depending on the virus [128,129]. This extended conformation is termed the pre-hairpin intermediate (PHI). Given the instability of the PHI, the HR1 and HR2 domains are subsequently driven to coalesce together, in the process bringing the transmembrane domains and fusion peptides into close proximity (Figure 6, Panel 3). Gradual lipid mixing between the bilayers will occur (Figure 6, Panel 4), starting with mixing of the outer leaflet lipids, which progresses to mixing of the inner leaflet lipids. The “zippering” to form the highly stable six helix bundle conformation (6HB), which is comprised of an inner trimeric core of HR1 α-helices intermingled with the HR2 α-helices at the interstices, ultimately merges the two membranes forming a fusion pore (Figure 6, Panel 5). This refolding of PMV-F from the metastable prefusion F conformation to the highly stable 6HB drives membrane fusion over a high-energy barrier, as lipid membranes do not spontaneously merge under physiological conditions. Thus, the F protein facilitates membrane fusion by inducing membrane stress and lowering the activation energy needed for the lipid merger. 

Inhibitory peptides that mimic the HR domains capture the conformational rearrangements in F. An HR2 domain-derived peptide halts fusion at the pre-hairpin intermediate (PHI) by binding to the HR1 trimer core and preventing the HR domains from coalescing to form the 6HB [130]. 

Brindley et al. (2014) demonstrated full closure of the 6HB is not necessary for MeV entry [131]. Covalent bonds were engineered into membrane-proximal positions of the HR2 domain that would be incompatible with the complete formation of a closed 6HB core. These mutant MeV-F proteins retained the ability to open productive fusion pores for viral nucleocapsid passage, as detected by cell-cell fusion and viral entry kinetics. It remains to be seen whether this lack of dependency on 6HB formation is specific to MeV or applicable to other PMVs. Nonetheless, the result has implications for the use of HR inhibitory peptides as antiviral therapeutics because it indicates inhibitory HR2 peptides may need to populate two or all three HR1 triple-helix grooves simultaneously in order to inhibit viral entry. 

### 4.1. PMV Fusion Protein Atomic Structures Reveal High Structural Conservation within the Family

The currently available atomic structures of PMV-Fs reveal high structural conservation, which is expected given the highly conserved mechanism of membrane fusion [92,94,95,96]. For example, the structure of the Hendra (HeV) virus F glycoprotein in its prefusion form revealed close structural similarity to pre-fusion PIV5-F (rmsd of 2.3 Å over 1216 equivalent Cα positions) [94] despite relatively low sequence similarity (~60%). One notable region of structural deviation was the FP cleavage site, which potentially reflects these two viruses’ recognition by different proteases during posttranslational processing. The cathepsin L substrate-binding pocket forms a longer, narrower groove than furin, which explains the presence of an elongated loop structure surrounding the cleavage site in HeV-F that is absent in the PIV5-F structure. This flexible loop is also seen in two distinct conformations in the structure of pre-fusion NiV-F [98].

Interestingly, electron tomographs of NiV virus-like particles supported a hexamer-of-trimers organization of pre-fusion NiV-F [95]. Crystallographic studies indicate that the interface between the trimers occurs at neighboring hydrophobic patches. These patches occur where the C-terminus of F_2_ and the N-terminus of the FP fold into a continuous *β*-sheet with the F_1_ subunit. This *β*-sheet conformation stabilizes the FP in its pre-fusion state. Disturbance of the hexameric assembly would disrupt these *β*-sheets, leading to triggering. Suitably, these residues forming the *β*-sheet are referred to as fusion “priming sites”—two of which associate with neighboring F trimers, while one putatively associates with the G protein, ultimately forming a hexamer of F trimers in a ring-like assembly. This provocative model suggests a single activating ephrin/NiV-G/F trimer interaction could ultimately result in the formation of eighteen 6HBs and membrane merger sites. It is currently unclear whether this arrangement is relevant to other PMV fusion proteins.

### 4.2. Novel Functions of the PMV-F Transmembrane Domain

Recent data suggests the transmembrane domain (TM) of the PMV-F proteins does more than just serve as a hydrophobic membrane anchor. It was originally observed that swapping the TM domains between NDV-F and two other related PMV-Fs, SeV-F and MeV-F, abolishes membrane fusion without affecting F cell surface expression [132]. 

Using analytical centrifugation, the isolated TM domains of HeV-F and PIV5-F were found to associate in monomer-trimer equilibria [133]. Given these two PMVs are distantly related, these results suggest trimeric TM-TM interactions are a common phenomenon for PMV-F proteins, and these interactions could have functional relevance in entry. Moreover, the TM domains of the different PMV-F proteins are not well conserved sequence-wise, indicating both sequence and structural determinants likely drive TM trimerization. 

The TM domain was subsequently shown to stabilize the F protein in its pre-fusion conformation [134]. A heptad repeat pattern of *β*-branched residues (e.g., leucine, isoleucine, threonine, valine) was identified to be highly conserved in the TM domains of 140 PMVs. Mutagenesis of the heptad repeat, leucine-isoleucine zipper (LIZ) in an isolated HeV-F TM domain resulted in a shift in propensity to form monomers concomitant with a complete loss of fusion activity. Using a conformational antibody specific for pre-fusion HeV-F, the HeV-F with a mutated LIZ TM domain also exhibited a drastic reduction in pre-fusion F at low temperatures unconducive for wild-type HeV-F triggering. Thus, TM-TM association may contribute to pre-fusion stability of the F protein, and a shift in equilibrium to a monomeric state destabilizes the pre-fusion conformation. Overall, it seems the F protein exists in a dynamic equilibrium—the TM-TM interactions serve to hold the F protein in its pre-fusion conformation until receptor binding occurs, whereupon conformational changes must occur within the TM domain that ultimately allow triggering.

Finally, co-expression of the isolated HeV-F TM domain with native, full-length HeV-F resulted in reduced expression of the native HeV-F and disrupted fusion activity, likely because interaction of the two forms leads to either protein misfolding during trafficking or premature triggering at the cell surface [135]. These effects are virus-specific, as the isolated HeV-F TM domain has no deleterious effect on native PIV5-F. Despite the low sequence conservation between the HeV-F and PIV5-F TM domains (26% sequence homology), they have both a disruptive effect on their respective native, full-length fusion proteins, indicating TM-TM interactions play a conserved role in PMV-F functions. In addition, small molecules that destabilize the TM domain could be attractive anti-HNV drug candidates.

## 5. The Paramyxovirus RBP Stalk Relays the Fusion Triggering Signal

PMV RBPs translate the input signal of receptor binding into the output result of membrane fusion. Two models (association and dissociation) describe how this process occurs [44,45,46] (Figure 7) for PMVs that use either SA- or protein-based receptors. The association or provocateur model [136,137] explains the triggering process for SA-using RBPs, such as PIV5, HPIV3, NDV, MuV, etc. (Figure 7A). These RBPs do not associate with or trigger their homologous fusion proteins at the cell surface until after receptor binding [104,138,139]. Protein-using RBPs, i.e., the MeV-H and HNV-G proteins, appear to follow the dissociation model, where the RBPs remain in complex with their fusion protein partners until receptor binding occurs (Figure 7B). Studies indicate the strength of the F-G or F-H interaction is inversely related to fusion activity, indicating that, for the protein-using PMVs, dissociation of F from the cognate RBP upon receptor-binding is necessary for membrane fusion [140,141,142,143]. The dissociation model has also been characterized as the “clamp” model in the literature [137], although the latter imputes a stabilizing role of the RBP on pre-fusion F for which there is little direct evidence. Regardless of the model, the RBP stalk domain appears to contain the F triggering signal, as headless versions of both SA- and protein-using PMV RBPs are sufficient to promote cell-cell fusion [103,144,145,146]. However, virions with these headless RBPs are non-infectious due to premature triggering of F before receptor binding occurs [144,145,146]. Overall, an “induced fit” hypothesis captures the conserved features of PMV entry mechanisms in the context of native, full-length RBPs [103]. This hypothesis states that any F and RBP (G/H/HN) interactions that occur during intracellular transport or at the cell membrane are non-triggering. Only upon stalk exposure, which is subsequent to receptor binding, does F activation occur. Although this stalk exposure mechanism is conserved amongst PMVs, the relative sites and flexibility of the F-activating regions differ (characteristics summarized in Table 1; models proposed in Figure 7; relevant residues in the context of structure are highlighted in Figure 4). 

### 5.1. Characteristics of RBP-HN for Triggering: A Central Hydrophobic Region Flanking the 4HB Kink

For closely related RBP-HN, hydrophobic residues in the stalk that immediately flank the kink region of the 4HB form the basis of F activation (Figure 4A; Figure 7A) [55,103,104]. The full ectodomain crystal structures reveal that these residues are normally shielded in the “heads down” conformation of the HN tetramer [55,102]. Structural flexibility in this region is required for triggering, as the addition of a single cross-linking cysteine (residue S92C) into NDV-RBP abrogates fusion triggering (Figure 4A; Figure 7A, Panel 3) [103]. Moreover, the addition of *N*-glycans in this region of the NDV-RBP stalk prevents both fusion and F-RBP complex formation without affecting RBP tetramerization, suggesting the F-interacting and F-activating regions are one and the same for SA-using RBPs (Figure 7A, Panels 3–4) [99]. Altogether, the central hydrophobic stalk region of HN RBPs provides both the specific transient interaction with and activating signal (“provocateur” model) to the homologous F proteins [104,147,148,149].

The addition of *N*-glycans above this central stalk region in PIV5-RBP disrupts fusion, but, as determined through mixed oligomer experiments with wild-type PIV5-RBP, acts recessively [88]. Furthermore, these upper stalk mutants do not affect oligomerization. Because the top of the PIV-RBP stalk accommodate these bulky *N*-glycans, these results further support the notion that a degree of structural flexibility exists at the top of the stalk. This flexibility likely facilitates the conformational change the heads undergo, from “heads down” to “heads up,” during fusion protein activation.

The crystal structures of the NDV-RBP and PIV5-RBP ectodomains provide further evidence for the “heads down” to “heads up” hypothesis for fusion protein triggering. The structure of the soluble NDV-RBP tetramer ectodomain in the absence of SA receptor revealed a four “heads down” conformation [55]. The lower head of each dimer interfaces with the 4HB. By mixing in a slight molar excess of the sialyllactose receptor, a structure of PIV5-HN ectodomain was generated that adopted a two “heads up” and two “heads down” conformation [102]. This structure supports the model of a dynamic head-stalk interaction, in which the globular head domain dimers form mobile structural units flexibly linked to the 4HB stalk. Disulfide bonds link the two monomers that are in identical orientations, i.e., the two “heads up” monomers form one dimer, while the two “heads down” monomers form the other. Two helices on one side of the 4HB stalk and a single globular head in the down position form a globular head/4HB stalk interface. The authors also explored the pliability of the tetramer by eliminating the native disulfide linkages and introducing disulfide bonds between one head in the up position and another in the down position. They refer to this novel interface as the dimer-of-dimers interface 2 (DOD_2_). The covalent linkages at this interface did not impair HN’s ability to activate fusion, despite the fact that this crosslinking should keep at least two of the globular heads in the “heads down” conformation. These results suggest that full formation of a four “heads up” conformation is not required for fusion protein-mediated membrane fusion.

A novel compound (CM9) that interacts with and causes RBP-HN to prematurely activate F was recently identified [156]. This compound is extremely potent, irreversibly triggering the viral fusion complex and resulting in nearly 100% reduction of HPIV3 infection upon preincubation with both laboratory-adapted and clinical isolate strains. Thus, this compound represents one of many of the potential antivirals that could be developed to exploit the “provocateur” model of fusion triggering and induce SA-using viruses to self-inactivate. 

### 5.2. Characteristics of RBP-G for Triggering: A Unique Proline-Rich Membrane Distal C-terminal Subdomain

The membrane distal C-terminal region (residues 159–167) of the NiV-RBP stalk is required for efficient fusion triggering (Figure 7B) [106]. The F-activating region of HNV-RBPs is located higher up in the stalk relative to HN or H RBPs (Figure 4B). Using conformational antibodies described later in this text [157], it was determined that this portion of the stalk allows receptor-induced conformational changes in the globular head to be relayed to the stalk [106,146]. This region is not required for F and G interactions, as a NiV-G deletion mutant lacking residues 146 to 182 immunoprecipitates the homotypic NiV-F [90]. The result suggests this stimulating region may only interact with NiV-F upon receptor binding. *N*-glycan additions in the lower stalk, more N-terminal (membrane proximal) to this F-activating region, severely impaired or completely eliminated fusion promotion, while retaining the RBP-F interaction [150]. These over-glycosylated, fusion-deficient stalk mutants did not exhibit enhanced binding to a monoclonal antibody (MAb) that interacts more strongly to NiV-RBP when pre-bound with (soluble) ephrin-B2 receptor. Therefore, despite still interacting with F, these mutants fail to undergo the conformational changes necessary for fusion triggering, i.e., the exposure of the C-terminal portion of the stalk. Thus, the F-interacting vs. F-activating domains of NiV-RBP are distinct. This is in contrast to RBP-HN/F (discussed in the previous section), where the stalk domain completely mediates the envelope protein interactions. Instead, a bidentate interaction exists between NiV-F and both the globular head and 4HB stalk of NiV-RBP (Figure 7B, Panel 1). Indeed, flow cytometric strategies with various truncated versions of soluble NiV-G and full length NiV-F expressed on cells confirmed this bidentate interaction [151]. This study also showed the soluble ectodomain version of NiV-G is not sufficient to induce cell-cell fusion, and only membrane-bound NiV-G triggers F. The data suggests anchoring of NiV-G is necessary for F activation or the transmembrane domain and/or cytoplasmic tail of G is involved in the fusion process. 

The F-activating residues in the C-terminal stalk region are part of a unique proline-rich subdomain that is 100% sequence conserved between the NiV-G and HeV-G stalks and only present in members of the HNV genus [90]. This subdomain also contains the cysteine residues that tetramerize the 4HB and that are required for fusion triggering, highlighting the importance of oligomeric stability in HNV entry [90] (Figure 4B; Figure 7B, Panel 2). In sum, the presence of the three disulfide linkages for tetramerization renders the RBP-G stalk less flexible than the stalk domains of RBP-HN or RBP-H, which are tetramerized by non-covalent interactions [48,55,85,86,88,90,101,153,154]. Thus, unlike the SA-using PMV RBPs, flexibility in the stalk domain must not be as important for HNV membrane fusion.

As mentioned above, a headless NiV-RBP containing residues 159–167 triggers cell-cell fusion without requiring receptor binding; however, virions containing the headless NiV-RBP do not enter cells, as NiV-F prematurely triggers before a target membrane is accessible [146]. Extensions beyond residue 167 were not as fusogenic. This data suggests regions past residue 167 could make the stalk too structurally stable for F triggering, and the signal that propagates from receptor binding in the context of the full length NiV-G ordinarily releases this structural constraint. Alternatively, the structures that fold over and cover the F triggering residues of the stalk domain may begin at residue 168. 

For NiV-G, two conformational monoclonal antibodies detect two distinct structural changes that occur after receptor binding: MAb 213 binding decreases after receptor binding (i.e., exhibits receptor binding reduction—RBR) (Figure 7B, Panel 2), whereas MAb 45 binding increases post receptor binding (i.e., exhibits receptor binding enhancement—RBE) (Figure 7B, Panel 3) [146,157]. MAb 213 binding does not compete with ephrin-B2 receptor binding to NiV-G, indicating that, instead of direct competition, the RBR is due to a conformational change that is disrupting the MAb 213 epitope (Figure 7B, Panel 2) [146]. Full length MAb/F(ab)_2_ competition studies between 213 and 45 show that, while pre-binding with a F(ab)_2_ 45 does not affect the RBR seen with MAb 213, the reciprocal order, i.e., pre-binding with a F(ab)_2_ 213, abrogates the RBE observed with MAb 45 [146]. This result indicates the conformational change detected by MAb 213 occurs prior to the change detected by MAb 45. MAb 213 binds to an epitope close to the ephrin-B2 receptor binding site, whereas MAb 45 binds to a region on a completely different face of the NiV-G monomer, directly C-terminal to the stalk domain and at the base of the head domain. The epitope of MAb 45, which is exposed upon receptor binding and putatively responsible for F activation, not surprisingly overlaps with the C-terminal region of the stalk implicated in NiV-F triggering [106]. 

Chimeric RBPs made between NDV-HN and NiV-G reveal the differences in the mechanisms of fusion protein activation [152]. A chimera possessing an NDV-HN-derived stalk and NiV-G-derived head effectively triggers NDV-F for fusion upon ephrin-B2 receptor binding. The result demonstrates that the stalk region defines the F-interactive and -activating site on HN. In contrast, the reciprocal chimera possessing a NiV-G-derived stalk and NDV-HN-derived head is incapable of complementing NiV-F. These results suggest the triggering cascade for NiV may be more complex than that of NDV because not only the stalk, but also the globular head contributes to NiV-F-induced fusion. However, the contribution of the globular head must only be essential in the context of full length NiV-G, as cell-cell fusion, but notably not virion entry, occurs when a headless NiV-G is used [146]. These results demonstrate the globular head acts as a gatekeeper to ensure NiV-F is only triggered when the virion is in close proximity to a target cell membrane, as sensed by receptor binding. 

### 5.3. Characteristics of RBP-H for Triggering: A Staggered-Head Alignment

Engineered N-glycans in the membrane proximal region of the MeV-RBP (H) stalk do not affect F/RBP-H complex formation or MeV-F triggering, whereas additions in the membrane distal region reduce glycoprotein interaction and completely block F triggering [108]. Specifically, residues 111, 114, and 118 in the central portion of the stalk were identified as modulators of glycoprotein interaction and determinants of F triggering (Figure 4C). Furthermore, this central region of the stalk (residues 84 to 118) needs to be positioned at a certain height relative to the donor/viral membrane, and, thus, to the neighboring prefusion F trimers, i.e., a staggered-head alignment must be present between MeV-F and MeV-H. Elongations introduced above residue 118, regardless of the length, have no impact on fusion activation, indicating that, similar to the HN RBPs, interactions between the MeV-RBP globular head and MeV-F are not essential for fusion triggering. 

Cysteines introduced in the membrane proximal region of the MeV-H stalk (residues 59–79) also do not exhibit deficiencies in fusion, further indicating the membrane proximal segment is not directly involved in fusion triggering [158]. In contrast, cysteines introduced in the central region of the stalk resulted in complete loss of fusion; however, the function of some of these mutants (residues 84–88 and 105–110) was rescued by reduction of the introduced disulfide linkages with DTT. This restoration of fusion activity indicates conformational flexibility, rather than the exact sequence, of this segment is important for efficient triggering. Indeed, in comparison to the highly stable, disulfide-tetramerized HNV-G stalk, the MeV-H stalk possesses more flexibility, as the disulfide-linked MeV-H dimers associate non-covalently into tetramers. However, some of the cysteine substitution mutants (residues 92–99 and 111–118) were unable to be rescued with dithiothreitol DTT treatment, demonstrating the importance of the amino acid sequence at those particular positions for fusion activation to occur.

The most C-terminal portion of the MeV-H stalk (termed the linker region, residues 140–154) contains a hydrophobic residue (I146) that is essential for F activation (Figure 4C) [109]. Mutation of this residue alters the disulfide linkages that occur through C139 and C154 in the upper stalk, resulting in the formation of non-productive, highly stable tetramers that fail to trigger fusion. Treatment with DTT does not fully restore fusion promotion activity, indicating that the inability to activate F is not solely caused by the presence of interdimer disulfide bridges. Thus, something specific to the isoleucine at this position or the overall structure and flexibility of this linker plays a modulatory role in F triggering activity. This linker forms a putative hydrophobic hinge region that connects the globular head to the 4HB. The flexibility of the head-to-stalk hydrophobic hinge region of the PIV5-HN protein was shown to be essential for triggering virus fusion [159]. The proposed mechanism involves the bulky head of HN swinging to-and-fro via this hinge region to facilitate timely HN-mediated F triggering. Based on the importance of residue I146 in fusion promotion activity, it appears the MeV-H linker behaves similarly.

## 6. Functions of Glycans: Henipavirus Strategies for Antibody Evasion and Maximal Host Survival

Glycan modifications on the HNV glycoproteins modulate fusion activation, protect the virus from antibody neutralization, and mediate interactions with cell surface expressed glycan-binding proteins. 

Six *N*-glycosylation sites were experimentally verified on NiV-RBP—one site in the stalk domain (termed G2) and five sites in the head domain (G3–G7) [160]. G2 in the stalk was shown to be critical for fusogenicity. In contrast, removal of *N*-glycans from the globular head of NiV-RBP results in hyperfusogenic phenotypes, suggesting the presence of *N*-glycans in the wild-type virus could impair cell-cell fusion levels in order to reduce syncytia formation and cell death, thus allowing maximum host survival and viral spread. Moreover, these sites play a role in glycan shielding, as removal of any of the single sites results in increased neutralization sensitivity to anti-NiV-G specific polyclonal antisera. HeV-RBP also contains these six highly conserved *N*-glycan sites, and there are similar trends in the effects of their removal [161]. 

Four *N*-glycosylation sites were experimentally and structurally verified on NiV-F [97,98,140,162]. Loss of the *N*-glycans results in enhanced membrane fusion and viral entry, and the removal of multiple N-glycans has a synergistic effect on fusion enhancement. Similar to the role *N*-glycans play for HNV-G, removal of the N-glycans increases sensitivity to antibody neutralization. Mechanistically, the hyperfusogenic F glycan mutants exhibited a faster transition to 6HB formation, as determined by increased resistance to an HR2 domain inhibitory peptide. The faster kinetics are likely due to decreased avidity to the NiV-RBP. 

The presence of *N*-glycans on NiV-F acts as a double-edged sword, as dimeric galectin-1 expressed on endothelial cells targets an *N*-glycan site on NiV-F and causes F to inappropriately oligomerize [163]. This forced crosslinking decreases the lateral mobility of NiV-F on the plasma membrane, prevents endocytosis and proper maturation of NiV-F, and inhibits the fusogenic activity of mature NiV-F [162]. However, this glycan binding also enhances NiV virion attachment to and infection of cells by bridging cognate viral and cell surface *N*-glycans [164]. The dual and opposing effects of galectin-1 are dictated by the timing of exposure to NiV-F. An enhancing effect occurs when galectin-1 binds during the initial phase of viral attachment to the target cell membrane, whereas an inhibitory effect occurs post-infection when galectin-1 targets NiV-F expressed on the cell membrane. 

Stone et al. (2016) explored the role of *O*-glycans in the HNV stalk [107]. These *O*-glycans were initially identified in HeV-RBP by mass spectrometry [165]. Alanine scanning mutagenesis was performed on the eleven identified *O*-glycan sites, and five of the mutants showed differences in cell-cell fusion. The hyperfusogenic mutants of the group all had decreased F association, supporting the dissociation model for HNVs. But *O*-glycan loss also affected cleaved F (F_1_) incorporation into pseudotyped virions. The data suggest the strength of the HNV-RBP and F interaction plays a role in F incorporation during budding. Interestingly, HNV *O*-glycan mutant pseudotyped virions produced at 32 °C instead of 37 °C regained proper F incorporation similar to those of wild type pseudotyped virions, suggesting that the lowered temperature allows for a more measured pace of F/RBP association, a process that was likely compromised by the *O*-glycan mutations. Overall, it seems like loss of the *O*-glycans decreases G/F avidity, which results in hyperfusogenic phenotypes in the context of triggering, but also results in lowered F incorporation. The authors also observed that, unlike *N*-glycans on HNV-G, the *O*-glycans provided no antibody neutralization shielding effects; however, the antisera they used might not have contained stalk-targeting antibodies. Interestingly, these *O*-glycan sites occur at a region of the stalk that is not predicted to fold into an α-helix, providing the putative kink in the HNV-G stalk (Figure 4B). It is also plausible that, if these *O*-glycans were taken into account, the HNV-G stalk would fold into a structure atypical of the other PMV RBP stalks. Moreover, the presence of glycans potentially prohibits crystallization of the stalk domain [166].

## 7. Structural and Functional Constraints on the Envelope Glycoproteins

Compared to other RNA viruses, paramyxoviruses exhibit a relative lack of antigenic drift [167], suggesting structural and functional constraints on the viral proteins. For the PMV envelope glycoproteins, the constraints on RBP and F of measles virus have been most well characterized. In vitro insertional mutagenesis experiments indicate F and RBP are relatively intolerant to insertions, which could explain the lack of antigenic diversity within the species [168]. Moreover, the F and RBPs amongst multiple MeV strains display low amino acid sequence variation, despite comparing strains that were isolated at various years and differentially passaged in cell culture [169,170,171,172,173].

More recently, Avanzato et al. (2019) identified a vulnerable neutralizing epitope on NiV-F targeted by multiple, independently produced monoclonal antibodies [98]. This epitope is highly conserved between HeV- and NiV-F. They next determined whether there was positive selection at this site, given that antibody targeting should drive evolutionary change. They also examined positive selection within the nucleotide sequences of MeV-F and PIV5, which share similar structural and functional features as NiV-F and for which there are more available sequences (e.g., 56 sequences for MeV-F, 26 sequences for PIV5-F, 18 sequences for NiV-F). They determined, within the sequences of all three viruses, locations that exhibit diversifying positive selection (i.e., rapidly changing amino acid sequence over evolutionary time) are rare. Notably, a positive selection signature was absent at the neutralizing epitope. The purifying negative selection present at the majority of sites and especially at this exposed region, despite the existing pressure from the host immune response, suggests strong functional and structural constraints on the fusion protein.

## 8. Insights from Sequence Conservation amongst the PMV-RBPs and PMV-Fs

We examined the sequence diversity of the RBPs and fusion proteins within the family by generating sequence-based phylogenies (Figure 8). We found the fusion proteins are more conserved than the RBPs (46% similarity amongst PMV-RBPs vs. 65% amongst PMV-Fs, on average) (Figure 8A), presumably because the PMV fusion proteins follow one common mechanism of action—to undergo the conformational changes required for membrane merger. Consequently, the proteins exhibit similar structural features (e.g., a hydrophobic fusion peptide, α-helical heptad repeats) reflected in the highly conserved sequences. In contrast, the RBPs have varying mechanisms (e.g., some possess NA activity, a wide variety of receptors are used, differential F triggering mechanisms, etc.), which has resulted in sequence divergence. 

We also investigated which domain, globular head or stalk, contributes more to the sequence conservation in the RBP. By examining intra-genera comparisons, we uncovered an intriguing difference between SA-using and protein-using PMVs. Amongst the SA-using PMVs, the head is more conserved when compared to the stalk (81% similarity in the heads vs. 72% similarity in the stalks, on average); contrastingly, for the ephrin-B2-using PMVs (66% similarity in the heads vs. 74% similarity in the stalks, on average) and CD150/Nectin4-using PMVs (57% similarity in the heads vs. 74% similarity in the stalks, on average), the stalk is more conserved (Figure 8B). These differences are significant (Figure 8C).

As a final test to this trend, we examined members of the *Pararubulavirus* genus. These viruses are genetically related to the SA-using orthorubulaviruses; however, they have either been experimentally shown [34] or are not predicted to use SA (Figure 2). We find these viruses reflect the same trend as protein-using PMVs (Figure 8C), further suggesting their use of protein receptors. 

Perhaps SA-using viruses have such low conservation in the stalk because the presence of specific residues is not required to trigger F, but rather the general biochemical character or a structural motif that can be generated from varying combinations of amino acids triggers F. In contrast, protein-using PMVs could require specific amino acids (e.g., the proline-rich region for HNV-RBP stalks) to occur at specific positions in the stalk for fusion activation to occur, hence the increased amino acid sequence conservation.

## 9. Conclusions

The central mechanism of PMV fusion activation is highly conserved. To avoid non-productive triggering events, the RBP acts as a regulatory switch to ensure fusion protein activation occurs only when the target and viral membranes are in close proximity. Moreover, the relative lack of antigenic drift in the envelope glycoproteins suggests structural and functional constraints will maintain this high conservation within the *Paramyxoviridae* family over evolutionary time. However, the specific features that enact the fusion cascade vary. Identifying these features advances the development of novel antivirals that impede PMV entry. Finally, the trends in conservation between SA-using vs. protein-using PMVs imply that, even if a pan-PMV antiviral is not possible, exploiting the commonalities in entry mechanisms will allow us to generate at least partially broad spectrum anti-HNV therapeutics. 

## Figures and Tables

**Figure 1 viruses-12-00161-f001:**
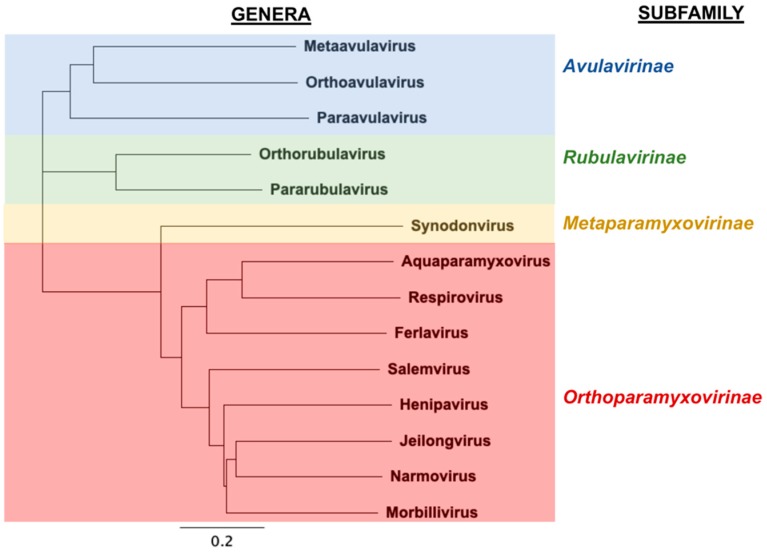
Phylogenetic tree of the family *Paramyxoviridae*. L protein sequences from representative species of the 14 genera—*Metaavulavirus* (APMV-2-L, YP_009513219.1), *Orthoavulavirus* (NDV-L, AAS67167.1), *Paraavulavirus* (APMV-3-L, YP_009094217.1), *Orthorubulavirus* (MuV-L, NP_054714.1), *Pararubulavirus* (SoRV-L, YP_009094034.1), *Synodonvirus* (WTLPV-L, AVM87369.1), *Aquaparamyxovirus* (ASPMV-L, YP_009094152.1), *Respirovirus* (HPIV3-L, ABY47607.1), *Ferlavirus* (FDLV-L, NP_899661.1), *Salemvirus* (SalV-L, YP_009094339.1), *Henipavirus* (NiV-L, NP_112028.1), *Jeilongvirus* (BeiV-L, YP_512254.1), *Narmovirus* (MossV-L, NP_958055.1), and *Morbillivirus* (MeV-L, NP_056924.1)—were aligned by ClustalW and a neighbor-joining tree was generated using Geneious Prime 2019.2. Scale bar of the resultant neighbor-joining tree indicates 0.2 amino acid substitutions per site. For clarity, only the parent genus of the representative species is indicated. Members in each subfamily cluster are shaded in blue (subfamily *Avulavirinae*), green (subfamily *Rubulavirinae*), yellow (subfamily *Metaparamyxovirinae*) and red (subfamily *Orthoparamyxovirinae*), respectively.

**Figure 2 viruses-12-00161-f002:**
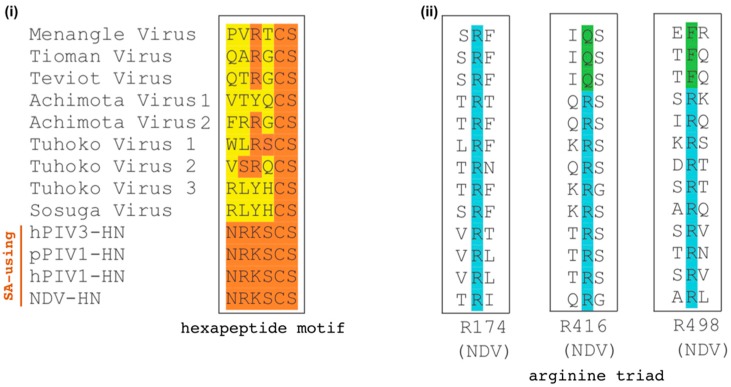
Parabulavirus receptor binding proteins (RBPs) lack motifs essential for binding and hydrolyzing sialic acid (SA). Hemagglutinin-neuraminidase (HN) proteins from known SA-using paramyxovirus (PMVs) human PIV3 (hPIV3-HN, NC_001796.2), porcine PIV-1 (pPIV1-HN, NC_025402.1), human PIV-1 (hPIV1-HN, NC_003461.1), and NDV (NDV-HN, NC_002617.1) were aligned with the RBP sequences from the genus *Pararubulavirus*, e.g., Menangle virus (NC_007620.1), Tioman virus (NC_004074.1), Teviot virus (NC_028233.1), Achimoto virus 1 (NC_025403.1), Achimoto virus 2 (NC_025404.1), Tuhoko virus 1 (NC_025410.1), Tuhoko virus 2 (NC_025348.1), Tuhoko virus 3 (NC_025350.1), and Sosuga virus (NC_025343.1) using ClustalW. Alignment reveals pararubulaviruses lack the conserved motifs required for sialic acid binding and neuraminidase functionality. These sequence motifs include: (i) the hexapeptide NRKSCS and (ii) a triarginyl cluster.

**Figure 3 viruses-12-00161-f003:**
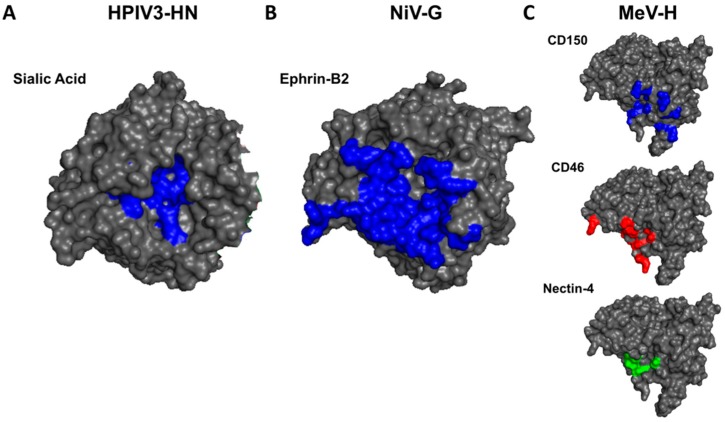
Receptor binding on MeV-H occurs on a different face of the *β*-propeller, as compared to the receptor binding sites on HPIV3-HN and NiV-G. Surface representations of HPIV3-HN, NiV-G, and MeV-H (PDB accession no. 1V3C, 3D11, 2RKC, respectively) are depicted with the monomeric head shown such that the center of the six-bladed *β*-propeller is pointing towards the reader. (**A**) Sialic acid binding residues on HPIV3-HN and (**B**) Ephrin-B2 binding residues on NiV-G are highlighted in blue. (**C**) top, CD150 binding residues on MeV-H are highlighted in blue; center, CD46 binding residues in red; bottom, Nectin-4 binding residues in green. The ligand-interface areas were determined by PDBePISA server [81].

**Figure 4 viruses-12-00161-f004:**
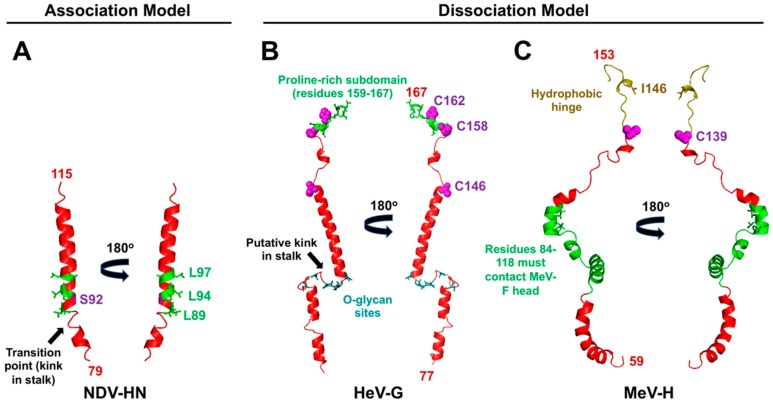
The F activation residues mapped onto the extant NDV-HN and modeled HeV-G and MeV-H stalk domain structures. (**A**) Two orientations of a cartoon representation of residues 79–115 in the NDV-HN stalk (PDB accession no. 3T1E) are shown in red. The hydrophobic residues shown as green sticks are those implicated in NDV-F activation [55,103,104]. A black arrow points to the transition point from the 7mer heptad repeat, supercoiled region to 11mer heptad repeat, straight region. Residue S92, colored in purple, signifies a position in the stalk where a disulfide linkage is not tolerated [103]. (**B**) Two orientations of a cartoon representation of the modeled HeV-G stalk domain (residues 77–167) are shown in red [105]. The proline-rich subdomain implicated in HNV-F activation is shown in green [106]. Cysteine residues involved in dimer-of-dimers formation are shown as magenta spheres [90]. The black arrow points to the putative helical transition site. O-glycosylated residues are shown as cyan sticks [107]. (**C**) Two orientations of a cartoon representation of the modeled MeV-H stalk region (residues 59–153) are shown in red [105]. Green residues are those implicated in MeV-F triggering [108]. Cysteine residue C139, involved in dimerization, is shown as a magenta sphere [109]. The hydrophobic hinge region that bridges the stalk to the head domain is colored olive, and residue I146, whose presence is essential for fusion, is shown as sticks [109].

**Figure 5 viruses-12-00161-f005:**
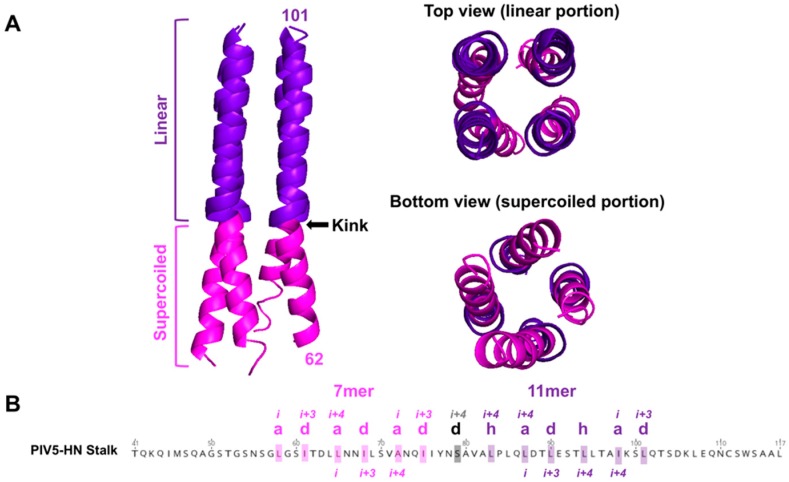
The tetrameric 4HB stalk exhibits transition from a supercoiled 7mer repeat lower region to a non-supercoiled, linear 11mer repeat upper region. (**A**) left, a side view of the PIV5-HN 4HB structure (PDB ID: 4JF7) is shown. The lower supercoiled portion is colored magenta. The upper linear region is colored purple. Arrow points to the transitional kink in the 4HB. Right, top view of 4HB stalk shows the 11mer linear organization, whereas bottom view depicts the supercoiling of the lower 7mer repeating monomers. (**B**) The annotated PIV5-HN stalk sequence is shown to highlight the heptad hydrophobic core repeats with the “a” and “d” positions (magenta) followed by the 11mer repeats with the “a,” “d,” and “h” positions (purple). S79 marks the polar kink point (black).

**Figure 6 viruses-12-00161-f006:**
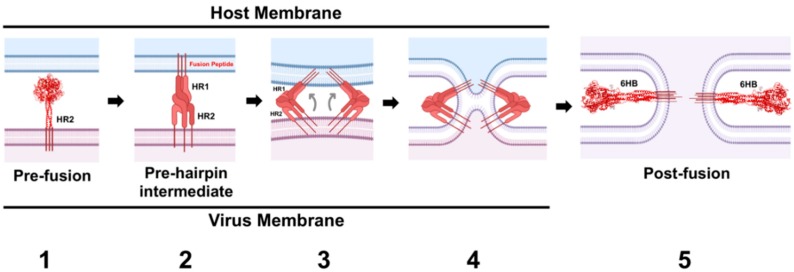
The PMV fusion protein facilitates merging of the virus and host membranes. The pre-fusion, cleaved HPIV3-F structure (PDB ID: 2B9B) is shown in red (**Panel 1**). Receptor-induced allosteric signals from the attachment protein induce release of the fusion peptide (FP), which inserts itself into the host cell membrane. Insertion of the fusion peptide coincides with the conversion of the HR1 domain from a set of compact helical structures into a highly unstable, extended helical trimeric coiled-coil domain. This transitional pre-hairpin intermediate (PHI) has not been crystallized, so a proposed schematic representation is shown. (**Panel 2**). The HR1 helices anchor F to the host cell membrane, while the instability of the PHI drives the HR2 alpha-helices to translocate 180° to pack against the interstices of the HR1 trimer (**Panel 3**). Gradual packing of the HR2 segments onto the trimeric HR1 core drives merging of the two lipid bilayers (**Panel 4**). Complete coalescing of the HR2 and HR1 segments marks the formation of the six-helix bundle (6HB) structure (the post-fusion NDV-F structure is shown in red, PDB ID: 3MAW), resulting in fusion pore formation and ending the process of membrane fusion (**Panel 5**). Schema was created with BioRender.

**Figure 7 viruses-12-00161-f007:**
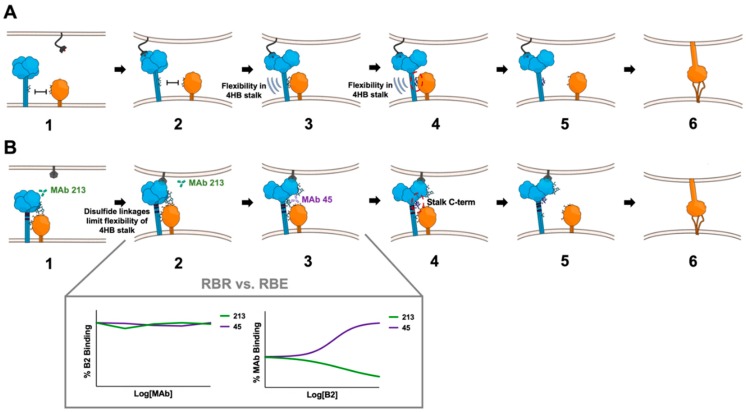
Models of the F activation pathways for SA- vs. B2-using PMVs. The PMV RBP (blue) and PMV F (orange) proteins are shown. Blue and orange sticks denote hypothesized regions of interacting residues but are not meant to represent specific amino acids. (**A**) Proposed model for SA-using PMVs. Prior to receptor binding, the HN tetramers and F trimers do not hetero-oligomerize because of the four “heads down” conformation (**Panel 1**). HN binds to sialic acid present on glycoproteins or glycolipids (**Panel 2**). Receptor binding leads to structural transition of the head domains to a “heads up” conformation, enabling F trimers to contact HN. The flexibility of the noncovalently-associated 4HB stalk facilitates the “heads up” transition, and stalk refolding exposes residues important for F interaction and triggering (**Panel 3**). The post-receptor-binding trigger-competent central stalk region sends a destabilization signal (dark blue residues, circled in red) to F (**Panel 4**). Relaying of the signal results in F and HN dissociation (**Panel 5**) and destabilization of the F trimers to form the pre-hairpin intermediate (PHI) (**Panel 6**). (**B**) Proposed model for B2-using PMVs. The G tetramers and F trimers associate via the globular head and 4HB stalk prior to receptor binding, i.e., a bidentate interaction. Conformational MAb 213 binds proximal to the B2 receptor binding site (**Panel 1**). Binding of the B2 receptor reduces MAb 213 binding—RBR (**Panel 2**). Receptor binding causes the tetrameric heads to transition to the “heads up” conformation, exposing a proline-rich subdomain in the very C-terminal stalk. This conformational change also increases exposure of the MAb 45 epitope at the base of the globular head enhancing MAb 45 binding—RBE (**Panel 3**). Inset contains representative data from Liu et al. (2013) to illustrate these receptor-induced behaviors: left, neither MAb 213 nor MAb 45 compete with receptor ephrin-B2 binding on NiV-G; right, following receptor ephrin-B2 binding, MAb 45 exhibits enhanced binding to NiV-G, whereas MAb 213 exhibits reduced binding. This revealed proline-rich subdomain at the stalk C-terminal region signals F to trigger (dark blue residues, circled in red) (**Panel 4**). The preassembled hetero-oligomeric F/G complexes dissociate (**Panel 5**) and the free, destabilized F transitions to the prehairpin structural intermediate (PHI) (**Panel 6**). The CD150/Nectin-4-using RBP model is not shown here to avoid redundancy, as it behaves as a hybrid of the two F activation pathways, i.e., follows the dissociation model, but the F-interacting and -activating regions are located centrally in the 4HB stalk. Figure schema courtesy of Griffin Haas.

**Figure 8 viruses-12-00161-f008:**
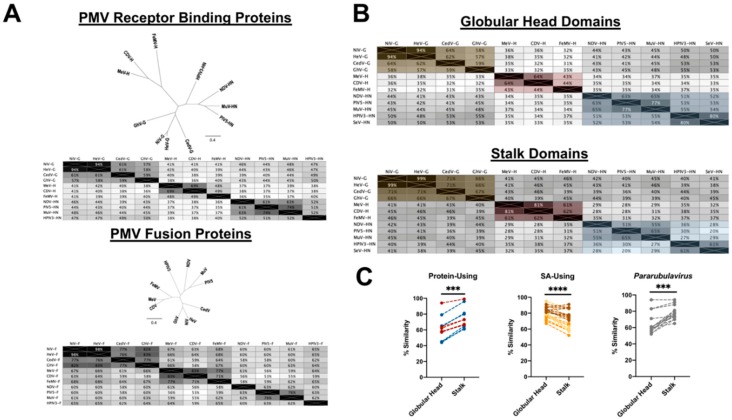
Sequence-based phylogenetic analysis of the PMV glycoproteins reveals glycan-using PMV RBPs have greater conservation in the head domain, whereas the protein-using PMV RBPs have greater sequence conservation in the stalk. Alignments, percent similarity matrices, and neighbor-joining trees were generated using Geneious Prime 2019.2. (**A**) Top, the full length sequences of NiV-G (NP_112027.1), HeV-G (NP_047112.2), CedV-G (YP_009094086.1), GhV-G (YP_009091838.1), MeV-H (P08362.1), CDV-H (NP_047206.1), FeMV-H (YP_009512963.1), NDV-HN (P12554.2), PIV5-HN (YP_138517.1), MuV-HN (AZM68937.1), and HPIV3-HN (P08492.1) were aligned by ClustalW. Scale bar of the neighbor-joining tree indicates 0.4 amino acid substitutions per site. The matrix contains percent similarity values that were calculated using a BLOSUM62 distance matrix with threshold = 0. Bottom, the full length sequences of NiV-F (AAV80428.1), HeV-F (NP_047111.2), CedV-F (YP_009094085.1), GhV-F (AFH96010.1), MeV-F (P69358.1), CDV-F (NP_047205.1), FeMV-F (YP_009512962.1), NDV-F (P12572.2), PIV5-F (P04849.1), MuV-F (AZM68935.1), and HPIV3-F (P06828.2) were aligned by ClustalW. Scale bar of the neighbor-joining tree indicates 0.4 amino acid substitutions per site. Matrix contains percent similarity values that were calculated using a BLOSUM62 distance matrix with threshold = 0. (**B**) Top, the isolated globular head domain amino acid sequences of the following eleven PMV RBPs were aligned—NiV-G (residues 176–602), HeV-G (residues 176–604), CedV-G (residues 199–622), GhV-G (residues 192–632), MeV-H (residues 175–617), CDV-H (residues 175–604), FeMV-H (residues 172–595), NDV-HN (residues 123–570), PIV5-HN (residues 118–565), MuV-HN (residues 133–582), HPIV3-HN (residues 141–572), SeV-HN (BAD74229.1; residues 143–575). The sequences were aligned by ClustalW, and the matrix shows percent similarity values that were calculated using a BLOSUM62 distance matrix with threshold = 0. Ephrin-B2-using PMV head comparisons are shaded yellow; CD150/Nectin-4-using PMV head comparisons are shaded red; SA-using PMV head comparisons are shaded blue. Bottom, the following isolated stalk domain amino acid sequences were aligned–NiV-G (residues 71–175), HeV-G (residues 71–175), CedV-G (residues 94–198), GhV-G (residues 88–191), MeV-H (residues 59–174), CDV-H (residues 59–174), FeMV-H (residues 53–171), NDV-HN (residues 46–122), PIV5-HN (residues 41–117), MuV-HN (residues 56–132), HPIV3-HN (residues 53–140), SeV-HN (residues 56–142). The sequences were aligned by ClustalW, and the matrix shows percent similarity values that were calculated using a BLOSUM62 distance matrix with threshold = 0. Ephrin-B2-using PMV head comparisons are shaded yellow; CD150/Nectin-4-using PMV head comparisons are shaded red; SA-using PMV head comparisons are shaded blue. (**C**) Comparison of sequence conservation in the globular head vs. stalk domains of protein-using PMVs (left), of SA-using PMVs (center), and of putative protein-using pararubulaviruses (right). Each data point represents a comparison between two species within the same genera (i.e., intra-genera comparisons). The percent similarity values for the sequence comparisons were plotted and paired (dotted line connects the pair) based on the region of the RBP being compared–the globular head or stalk. Comparisons between members of genus *Henipavirus* are shown in red (*n* = 6 paired comparisons), *Morbillivirus* in blue (*n* = 6 paired comparisons), *Orthoavulavirus* in brown (*n* = 10 paired comparisons), *Orthorubulavirus* in orange (*n* = 10 paired comparisons), *Respirovirus* in yellow (*n* = 6 paired comparisons). Globular head and stalk amino acid sequences used include those listed above and the following additional sequences: for genus *Morbillivirus*, RPV-H (YP_087125.2; head = residues 175–609, stalk = residues 59–174); for genus *Orthoavulavirus*, APMV9-HN (YP_009094363.1; head = residues 123–579, stalk = residues 46–122), APMV16-HN (YP_009508503.1; head = residues 123–618, stalk = residues 46–122), APMV12-HN (YP_009094172.1; head = residues 123–614, stalk = residues 46–122), APMV13-HN (YP_009255225.1; head = residues 123–610, stalk = residues 46–122); for genus *Orthorubulavirus*, HPIV2-HN (NP_598405.1; head = residues 123–571, stalk = residues 46–122), HPIV4-HN (YP_008378664.1; head = residues 128–574, stalk = residues 49–127), SV41-HN (YP_138509.1; head = residues 119–568, stalk = residues 42–118); for genus *Respirovirus*, BPIV3-HN (NP_037645.1; head = residues 141–572, stalk = residues 53–140), HPIV1-HN (NP_604441.1; head = residues 143–575, stalk = residues 56–142). A two-tailed, paired *t*-test (Wilcoxon matched-pairs signed rank test) was performed to detect a significant difference between the conservation in the globular head vs. stalk. *** denotes *p* < 0.001 and **** denotes *p* < 0.0001. To generate the percent similarity values for pararubulaviruses (*n* = 15 paired comparisons), isolated putative globular head domain amino acid sequences of representative members of the genus *Pararubulavirus* were aligned—SoRV-HN (YP_009094033.1, residues 135–582), MenV-HN (YP_009512970.1, residues 144–595), AchiV-1-HN (YP_009094457.1, residues 145–595), TeV-HN (YP_009512977.1, residues 144–595), TioV-HN (NP_665870.1, residues 144–593), TuV-1-HN (YP_009094497.1, residues 131–580). The sequences were aligned by ClustalW, and percent similarity values that were calculated using a BLOSUM62 distance matrix with threshold = 0. Moreover, the isolated putative stalk domain amino acid sequences of representative members of the genus *Pararubulavirus* were aligned—SoRV-HN (residues 57–134), MenV-HN (67–143), AchiV-1-HN (68–144), TeV-HN (67–143), TioV-HN (67–143), TuV-1-HN (54–130). The sequences were aligned by ClustalW, and percent similarity values were calculated using a BLOSUM62 distance matrix with threshold = 0.

**Table 1 viruses-12-00161-t001:** Comparison of fusion promotion mechanisms for PMV Receptor Binding Proteins.

	Ephrin-B2-Using RBP ^1,2^	SA-Using RBP ^3^	CD150/Nectin-4-Using RBP ^4^
**Effects of carbohydrate shielding along the stalk**	Addition of *N*-glycans along stalk retains RBP(G)-F interaction, but loss of fusion activity occurs [150]	Addition of N-glycans prevents both fusion and F-RBP(HN) complex formation [99]	Engineered *N*-glycans in the membrane-distal, but not the membrane-proximal region of the MeV RBP (H) stalk reduce F/RBP interactions and completely block F triggering [108]
**F-activating regions in stalk**	F-activating regions are located at the very C-terminal portion of stalk (residues 159–167) [106]	F-activating regions are located at central portion of stalk near the transition from the supercoiled 7mer repeat region to the linear coiled-coil 11mer repeat region (Figure 4) [55,103,104]	F-activating regions are located at a membrane distal region of the stalk (residues 84 to 118) [108]
**Characteristics of F/RBP interaction in triggering F**	Interaction is bidendate, involves head and stalk [151]; in the context of full length NiV-G, both head and stalk are necessary for productive NiV-F triggering [152]	Interaction and triggering occur through the stalk domain [104,147,148,149]	The C-term of the stalk can be elongated by 41-residues, adding a pitch of ~75 Å above residue 118, and triggering will still occur [108]; results suggest that interaction between MeV-H globular head and MeV-F is not necessary for fusion
**Truncated “headless” RBP**	Triggers fusion [146]	Triggers fusion [103,144]	Triggers fusion [145]
**RBP oligomerization/4HB flexibility**	Most rigidly structured stalk of PMVs; three cysteines (C146, C158, C162 in NiV/HeV) mediate dimerization and tetramerization of G; failure to tetramerize leads to reduced fusion [90]	Dimerization occurs via a disulfide linkage [48,86,101] and tetramerization occurs via non-covalent interactions in the stalk; weaker non-covalent interactions allow for flexibility of 4HB that is necessary for fusion [55,102]	Disulfide-linked dimers may associate non-covalently into a tetramer [85,153,154]; covalent tetramerization disrupts fusion triggering function, indicating some degree of flexibility is needed for fusion promotion [155]

^1^ Receptor binding protein (RBP) is the current ICTV nomenclature for paramyxovirus attachment glycoproteins, formerly designated variously as HN, H, or G. ^2^
**Henipavirus RBP** (subfamily *Orthoparamyxovirinae*, genus *Henipavirus*) such as those from NiV and HeV formerly designated as **NiV-G** and **HeV-G** all use at least ephrin-B2 (EFNB2) as their entry receptor. ^3^ Sialic acid (SA)-using RBP from major subfamilies and genera within the *Paramyxoviridae* such as **APMV-1 RBP**, including the formerly designated **NDV-HN** (subfamily *Avulavirinae*, genus *Orthoavulavirus*), **PIV5 and MuV RBP** formerly designated as **PIV5-HN** and **MuV-HN**, respectively (subfamily *Rubulavirinae*, genus *Orthorubulavirus*), and **HPIV3 RBP** formerly designated as **HPIV3-HN** (subfamily *Orthoparamyxovirinae*, genus *Respirovirus*). ^4^ Morbillivirus RBP (subfamily *Orthoparamyxovirinae*, genus *Morbillivirus*) formerly designated as **MeV-H**, **CDV-H**, etc. all use species-specific homologs of SLAMF1 (Signaling Lymphocytic Activation Molecule Family Member 1 aka CD150 or “SLAM” as often used in the literature to refer to this morbillivirus receptor) and Nectin-4 (alias PVRL4) as cell entry receptors. CD150 is the entry receptor on cognate myeloid and lymphoid cells when morbilliviruses initially enter and spread systemically within the host, while Nectin-4 is the epithelial cell receptor, which the virus uses to exit the host via the respiratory system.

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
