# Peer review of "Differential Features of Fusion Activation within the Paramyxoviridae"

_viruses, 2020, doi:10.3390/v12020161_

Round 1

Reviewer 1 Report

The authors wrote a good review of PMV entry pathway, focusing on the functional differences in fusion activation between members of the family. But some minor changes need to be made. I listed some examples as below.

Minor points:

Line 27: grammatical error for the sentence “comprise” should be changed to “comprises”. Line 42: grammatical error for the sentence “have” should be changed to “has”. Line 43: grammatical error for the sentence “have” should be changed to “has”. Line 64-70: “HN, hemagglutinin-neuraminidase, proteins bind sialic acid (SA) on red blood cells ……experimentally confirmed or predicted to bind proteinaceous receptors.” The authors are encouraged to add citations in this part. Line 170: the authors have already provided acronym of receptor binding protein at line 100, there is no need to do it again here.

Author Response

Response to Reviewer 1 Comments

The authors wrote a good review of PMV entry pathways, focusing on the functional differences in fusion activation between members of the family. But some minor changes need to be made. I listed some examples as below.

Point 1 (grammatical errors): Line 27: grammatical error for the sentence “comprise” should be changed to “comprises”. Line 42: grammatical error for the sentence “have” should be changed to “has”. Line 43: grammatical error for the sentence “have” should be changed to “has”.

 Response 1: We thank the reviewer for catching our grammatical errors. We have corrected the errors noted above and have carefully inspected the manuscript to catch the following additional errors - Line 75: “have” was changed to “has”; Line 149 “underscores” was changed to “underscore”; Line 152: “use” was changed to “uses”; Line 159: “suggests” was changed to “suggest”; Table 1, row 2, column 2: “retain” was changed to “retains”; Line 734: “are” was changed to “is”.

Point 2 (description of the RBP types): Line 64-70: “HN, hemagglutinin-neuraminidase, proteins bind sialic acid (SA) on red blood cells …experimentally confirmed or predicted to bind proteinaceous receptors.” The authors are encouraged to add citations in this part.

Response 2: We agree that citations should be added to this section describing the different types of RBP. We have added references 16-21, which originally report the hemagglutination and neuraminidase activities of SeV-RBP, NDV-RBP, PIV5-RBP, and HPIV3-RBP, to line 68. We have added reference 22, which is a 1991 review on the interactions of MeV-RBP with erythrocytes and cell lines, to line 70. We have added references 28-33, which describe the receptor binding activities of G-type RBPs, to line 74.

Point 3 (redundancy in title): Line 170: the authors have already provided acronym of receptor binding protein at line 100, there is no need to do it again here.

Response 3: We thank the reviewer for pointing out this redundancy. It has been corrected (line 178).

Reviewer 2 Report

Azarm and Lee wrote an excellent review on fusion activation by the glycoproteins of the Paramyxoviridae family. Membrane fusion is key to viral entry into host cells and on the pathognomonic cell-cell fusion characteristic of paramyxoviral infections. This review is thus not only quite significant, but also comprehensive and well written. The figures made for the review are all very clear and helpful to understand the concepts brought up. I only have few minor suggestions for meant for further improvement.

In line 68, the authors may want to add CD150 and Nectin 4 in addition to CD46, particularly since the first two are the most physiologically relevant.

Add references to support the statement in the sentence in lines 81-82, regarding spread of PMVs either via cell-free or cell-cell spread.

In the Figure legend of Fig. 4, sentence in lines 204-205, add references at the end of the sentence regarding the proline-rich subdomain in HNV-F, to be consistent with other sentences in the legend. Similarly, add references at the end of the sentence on the headless RBPs on virions being not infectious due to premature triggering of F after sentence in lines 372-373. Similarly, in Table 1, some references are there and some are missing in some parts of the table.

Author Response

Response to Reviewer 2 Comments

Azarm and Lee wrote an excellent review on fusion activation by the glycoproteins of the Paramyxoviridae family. Membrane fusion is key to viral entry into host cells and on the pathognomonic cell-cell fusion characteristic of paramyxoviral infections. This review is thus not only quite significant, but also comprehensive and well written. The figures made for this review are all very clear and helpful to understand the concepts brought up. I only have a few minor suggestions meant for further improvement.

Point 1 (addition of physiologically relevant  MeV-RBP receptors): In line 68, the authors may want to add CD150 and Nectin 4 in addition to CD46, particularly since the first two are the most physiologically relevant.

 Response 1: We agree with the reviewer that the more physiologically relevant MeV-RBP receptors should also be noted in this section. Accordingly, we have added the following statement in lines 70-72: “For systemic spread and respiratory transmission in humans, MeV-RBP uses the physiologically relevant SLAM (or CD150) and Nectin-4 proteins, respectively [23-27].” 

Point 2 (refs describing PMV spread): Add references to support the statement in the sentence in lines 81-82, regarding spread of PMVs either via cell-free or cell-cell spread.

Response 2: As requested, we have added references 41-43 to support the statement that PMVs can spread either cell-free or direct cell-to-cell [line 88].

 Point 3 (refs for Figure 4 legend): In the Figure legend of Fig. 4, sentence in lines 204-205, add references at the end of the sentence regarding the proline-rich subdomain in HNV-G, to be consistent with other sentences in the legend.

Response 3: Reference 106 has been added to the Figure 4 legend [line 215].

Point 4 (refs for headless PMV-RBPs): Similarly, add references at the end of the sentence on the headless RBPs on virions being not infectious due to premature triggering of F after sentence in lines 372-373.

 Response 4: References 144-146 have now been cited, which report the absence of infectivity for virions that possess headless RBPs [line 384].

 Point 5 (refs for Table 1): Similarly, in Table 1, some references are there and some are missing in some parts of the table.

Response 5: Appropriate references (e.g. [106], [55,103,104], [108]) have been added to the table for completeness.